# Daratumumab in systemic lupus erythematosus: a single-arm phase 2 trial

Lennard Ostendorf [1,2,3,4], Jan Zernicke[5], Jens Klotsche[1], Robin Kempkens[1], Anne E. Beenken[1,5], Robert Biesen [5], Qingyu Cheng[1,5], Laleh Khodadadi [1,5], Gabriela Maria Guerra[1], Frederik Heinrich [1], Pawel Durek [1], Gerd-Rüdiger Burmester[5], Gerhard Krönke [1,5], Falk Hiepe [1,5], Mir-Farzin Mashreghi [1,6] & Tobias Alexander [1,5] ✉

Antibody-secreting cells (ASCs) play a central role in the pathophysiology of systemic lupus erythematosus (SLE). This single-arm, open-label, phase 2 clinical trial aims to evaluate the safety and efficacy of the ASC-depleting anti-CD38 monoclonal antibody daratumumab in patients with SLE (NCT04810754). The primary endpoint is the reduction in serum anti-double-stranded DNA (anti-dsDNA) antibody levels at week 12. Key secondary end points include safety, clinical efficacy, and immunologic changes. Ten female patients with active disease and inadequate responses to at least two immunosuppressive drugs have received eight subcutaneous injections of 1800 mg daratumumab weekly, with dexamethasone as premedication (20 mg for first two injections, then 10 mg). By week 12, anti-dsDNA antibody levels have been reduced by a median of 109.6 IU/ml (95% CI 38.1 – 274.5). The treatment resulted in rapid and sustained clinical improvements across all patients and organ domains, reflected by a 100% SRI-4 (Systemic Lupus Erythematosus Responder Index-4) response rate at week 12. Hypogammaglobulinemia occurred in 5/10 patients, requiring immunoglobulin substitution. Daratumumab treatment has depleted circulating ASCs, reduced type I interferon activity, and profoundly modulated the T-cell responses. These findings highlight the pivotal role of ASCs in SLE pathogenesis and support daratumumab as therapeutic option for SLE.

In systemic lupus erythematosus (SLE), antibodies against a variety of nuclear antigens cause multi-organ inflammation and damage by deposition of immune complexes, induction of a type-I interferon (IFN) responses and direct autoantibody-mediated effects[1]. Such autoantibodies are secreted both by circulating antibody secreting cells (ASCs) and long-lived plasma cells (LLPCs), the latter residing in dedicated survival niches in the bone marrow[2]. In active SLE, circulating ASCs are more abundant and enriched in phenotypically mature

[1]German Rheumatology Research Centre (DRFZ), an Institute of the Leibniz Association, Berlin, Germany. [2]Department of Nephrology and Medical Intensive Care, Charité - Universitätsmedizin Berlin, corporate member of Freie Universität Berlin, Humboldt-Universität zu Berlin, and Berlin Institute of Health, Berlin, Germany. [3]BIH Biomedical Innovation Academy, BIH Charité Junior Clinician Scientist Program, Berlin Institute of Health at Charité - Universitätsmedizin Berlin, Berlin, Germany. [4]Division of Rheumatology, Inflammation, Immunity, Brigham and Women's Hospital, Harvard Medical School, Boston, MA, USA. [5]Department of Rheumatology and Medical Immunology, Charité - Universitätsmedizin Berlin, corporate member of Freie Universität Berlin, Humboldt-Universität zu Berlin, and Berlin Institute of Health, Berlin, Germany. [6]German Center for Child and Adolescent Health (DZKJ), Berlin, Germany. ✉e-mail: tobias.alexander@charite.de

ASCs[3,4], similar to bone marrow LLPC that originated from extra-follicular immune reactions[5]. LLPC are generated early and continuously in SLE[6] and may contribute to the refractoriness of the disease, as they are unresponsive to standard immunosuppressive therapies and CD20-targeting biologic disease-modifying drugs[7,8]. LLPCs represent a major compartment of immunologic memory driving chronic inflammation[9], but their targeting represents a challenge[10]. The therapeutic relevance of depleting LLPCs in SLE is supported by mechanistic mouse studies[11,12] and observations in SLE patients treated with immunoablation followed by autologous haematopoietic stem cell transplantation[13], use of the proteasome inhibitor bortezomib[14] and the CD3xBCMA bispecific antibody teclistamab[15,16]. However, these approaches are associated with significant side effects.

Targeting CD38, which is highly expressed on ASCs, recently emerged as a promising alternative approach for depleting plasma cell. The CD38 human monoclonal antibody daratumumab is effective in plasma cell killing and approved for the treatment of multiple myeloma[17,18]. Its use in refractory cases of SLE and other systemic inflammatory autoimmune diseases has resulted in encouraging clinical responses by depleting autoreactive ASC[19–21]. CD38 is both a transmembrane ecto-enzyme and a receptor for CD31 and its expression has been reported to be increased in several immune cell subsets in autoimmune diseases including SLE[22]. Enzymatically, CD38 degrades both extracellular and intracellular $NAD^+$ [23]. This results in a number of metabolic and epigenetic changes in T cells, at least in part through the decreased activity of Sirtuin-1[24]. Reduced $NAD^+$ levels lead to mitochondrial dysfunction, decreased cytotoxic function of $CD8^+$ T cells and decreased cytokine production in $CD4^+$ T cells[24–28], further supporting the potential therapeutic benefit of targeting CD38 in SLE by restoring normal T cell function.

In this phase 2 clinical proof-of-concept trial, patients with moderate-to-severe SLE and an inadequate response to at least two prior immunosuppressive/-modulatory drugs receive eight weekly subcutaneous doses of 1800 mg daratumumab plus dexamethasone as an add-on to stable background medication. Patients are monitored for 36 weeks. The primary objective is to evaluate the reduction in anti-double-stranded DNA antibody levels, as they reflect a measure of immunological disease activity and are primarily produced by $CD38^{high}$ ASCs and LLPCs, the main target of daratumumab. Secondary objectives include assessments of clinical efficacy, safety, pharmacokinetics, and immunological changes during treatment.

## Results

### Patients and treatment
Ten patients were enrolled between August 2021 and January 2023 (Fig. 1). Detailed baseline characteristics of enrolled participants are summarised in Tables 1–2. The median age was 38 (range 24-43) years. All patients had elevated anti-dsDNA antibodies in serum and active disease at baseline, reflected by a median Systemic Lupus Erythematosus Disease Activity Index 2000 (SLEDAI-2K, a weighted score of lupus activity across organ domains that ranges from 0 to 105) of 12 (range 8–20), despite a median 6 previous lines of treatment (range 2–9), including B cell-targeting therapies in 6 patients. All patients had musculoskeletal, mucocutaneous and haematologic manifestations and 90% had renal involvement in their disease history (2 with class II, 4 with class IV, including 1 with class IV/V, 2 with class V and 1 without kidney biopsy information). At baseline, persistent disease activity—defined as activity of grade C or higher according to the British Isles Lupus Assessment Group Index 2004 (BILAG-2004)—was present in 100% of patients in the mucocutaneous domain, 70% in the musculoskeletal domain, 60% in the renal and hematologic domains, and 10% in the neuropsychiatric domain (Supplementary Table 1). Except for one patient (Patient #8) missing the last two daratumumab doses due

to SARS-CoV2-infection, all patients received eight weekly doses of 1800 mg subcutaneous daratumumab in addition to dexamethasone (20 mg for the first two doses, 10 mg for all subsequent doses). All patients completed the trial. Background medication was not changed, while oral glucocorticoids could be tapered from week 12 at the discretion of the treating physician.

### Significant reduction of autoantibodies
Treatment with daratumumab and dexamethasone resulted in a rapid decline in serum anti-dsDNA antibodies in all patients, with a median reduction from 166.3 IU/ml at baseline to 61.1 IU/ml at week 12 (median difference −109.6, 95% confidence interval (CI): −274.5 – −38.1), i.e. 4 weeks after the last daratumumab injection, meeting the primary endpoint (Fig. 2A, Table 2). During follow-up, anti-dsDNA antibodies did not rise significantly again, although six patients had higher levels at the final visit compared to week 12. Analysis of antibodies to extractable nuclear antigens (ENA) showed mixed results. While Smith antigen (Sm)-directed antibodies declined in both patients with elevated levels at baseline, reduction in anti-SSA and anti-U1RNP antibodies was evident only in two out of five and six patients, respectively (Supplementary Fig. 1A). Serum complement levels for C3 increased from a median 875 mg/l to 955 mg/l at week 12 (median difference 130 mg/l, 95% CI 80 – 230), with a trend towards further improvement during follow-up (Fig. 2B).

### Clinical improvement in response to daratumumab
Reduction of anti-dsDNA antibodies during the treatment with daratumumab and dexamethasone was accompanied by a rapid and marked clinical improvement in all patients with a significant reduction of the SLEDAI-2K score from a median 12 at baseline to 4 at week 12 (median difference: -8, 95% CI: -6 - -10), which remained stable until the final study visit at week 36 (Fig. 2C, Table 2). Notably, the treatment led to clinical improvements in all major organ systems, particularly for joint and skin manifestations. This was reflected by reductions of the Clinical Disease Activity Index (CDAI, a score of arthritis severity that ranges from 0 to 76, typically used to assess disease activity in rheumatoid arthritis) from a median 11.5 to 0 (median difference -11.5, 95% CI: -13.5 – -4.0) as well as the Cutaneous Lupus Disease Area and Severity Index (CLASI-A, a score of cutaneous lupus manifestations that ranges from 0 to 76, with higher scores indicating greater severity), from 6.0 to 0 at week 12 (median difference -5.0, 95% CI: -6.0 – -3.0) (Fig. 2D and Supplementary Fig. 1B). In six patients with active lupus nephritis at baseline, defined as having a British Isles Lupus Assessment Group (BILAG) renal score of at least C, proteinuria reduced from a median 649 mg/g creatinine at baseline to 302 mg/g at week 12 (median difference -269 mg/g, 95% CI: -690 – -84). Overall, these responses translated into a SLE Responder Index (SRI)-4 response of 100% and 70% of participants at week 12 and week 36, respectively (Fig. 2E), despite a significant reduction of the median daily prednisolone dose from 6.25 mg at week 12 to 5.0 mg at week 36 (median difference -2.75, 95% CI: -3.5 – 0) (Fig. 2F). Remission according to DORIS (Definition Of Remission In SLE) criteria[29], defined as having a clinical SLEDAI-2K of 0, a Physician's Global Assessment (PGA) < 0.5 and prednisolone ≤ 5 mg/day, was achieved in 50% of participants at the final study visit. Two disease flares occurred at week 20 and week 24, defined as at least 1 new SLEDAI-2K mild/moderate or severe flare compared with baseline as assessed using the SELENA-SLEDAI flare index (SFI). Flares developed in patients #3 and #4 with symptoms similar to those prior to treatment, including joint and skin manifestations, resulting in the initiation of belimumab therapy. Finally, the health-related quality of life of all patients markedly improved under the treatment with daratumumab and dexamethasone, reflected by a significant increase in the Functional Assessment of Chronic Illness Therapy (FACIT-F) score, a 40-item measure that assesses self-reported fatigue, and in the SF-36-score, a

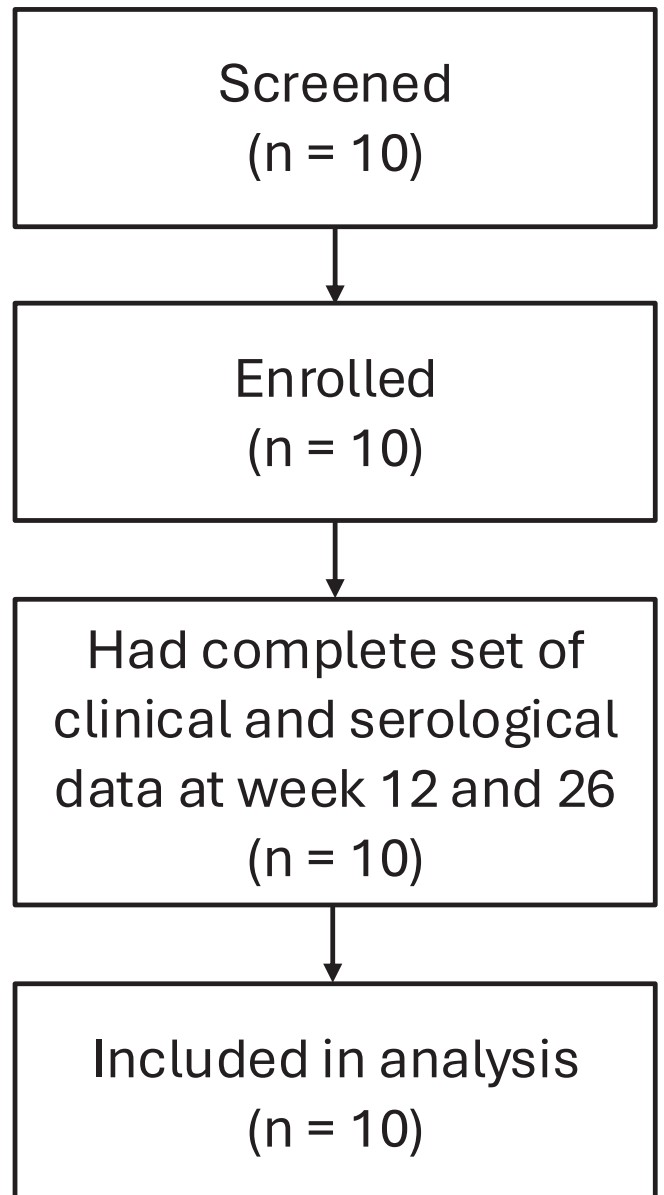

**Fig. 1 | CONSORT Chart.** Patient deposition in the trial.

36-item patient-reported questionnaire that covers eight health domains (Fig. 2G, Table 2).

**Safety and adverse events**

Treatment-emergent adverse events (TEAEs) observed are listed in Table 3. There were no severe adverse events or TEAEs leading to study discontinuation and the adverse event profile was similar to that reported for daratumumab monotherapy in multiple myeloma[17]. TEAEs occurred in nine out of ten patients, most frequently including infections, gastrointestinal events, and hypogammaglobulinemia. Immunoglobulin G (IgG) levels significantly declined at week 12 (median difference -6.9 g/dl, 95% CI: -8.4 – -3.2), (Fig. 2B) with values dropping below 5 g/l in five patients (Supplementary Fig. 1C). Similarly, vaccine-induced antibody titres for tetanus toxoid (TT) declined by 49%, suggesting a depletion of LLPCs secreting these antibodies (Table 2). Accordingly, five participants received a total nine prophylactic intravenous immunoglobulin (IVIG) infusions of 30 g, initiated at IgG levels below 5 g/l. A predictor for the development of hypogammaglobulinemia (IgG <5 g/l) was an initial IgG level of 11.5 g/l or below (Supplementary Table 2 **and** Fig. 2B). SARS-CoV2 infections

**Table 1 | Demographics and baseline characteristics of study participants**

| Pat | Age/Sex | Race | SLEDAI-2K baseline | Organ manifestations | Previous therapies* | Failed previous immunosuppressants | Continued immunosuppressants |
|---|---|---|---|---|---|---|---|
| 1 | 38/f | White | 12 | Arthritis, skin, alopecia, pleuritis, fever | 9 | HCQ, AZA, MTX, BEL, MMF, CsA, CYC, BAR, IVIG | HCQ, Pred 5 mg |
| 2 | 24/f | White | 20 | Renal (LN V), arthritis, skin, alopecia, pleuritis, hematologic, fever | 2 | HCQ, MFA | HCQ, MFA, Pred 7.5 mg |
| 3 | 40/f | White | 12 | Renal (LN V), arthritis, skin, alopecia, hematologic, pericarditis | 5 | HCQ, AZA, MMF, MFA, BEL | MFA, Pred 5 mg |
| 4 | 32/f | White | 10 | Renal (LN II), arthritis, skin, alopecia, hematologic | 8 | HCQ, AZA, MTX, BEL, BAR, RTX, IVIG, CYC | HCQ, Pred 12.5 mg |
| 5 | 35/f | Black | 10 | Renal (LN IV/V), arthritis, alopecia, rash, hematologic | 3 | HCQ, AZA, MMF | HCQ, MMF, Pred 5 mg |
| 6 | 27/f | White | 16 | Renal (LN IV), CNS, rash, alopecia, hematologic, arthritis | 8 | HCQ, MTX, AZA, CYC, MMF, RTX, BEL, IVIG | HCQ, Pred 5 mg |
| 7 | 43/f | White | 10 | Renal (no biopsy), arthritis, rash, alopecia, hematologic, | 7 | HCQ, AZA, MTX, CsA, CYC, MMF, BEL | Pred 9 mg |
| 8 | 41/f | White | 12 | Renal (LN IV), rash, ulcer, alopecia, arthritis, hematologic | 6 | HCQ, AZA, MTX, CYC, MMF, BEL | MMF, HCQ, Pred 5 mg |
| 9 | 38/f | Black | 8 | Renal (LN II), rash, alopecia, hematologic, arthritis | 2 | HCQ, AZA | HCQ, AZA, Pred 7.5 mg |
| 10 | 42/f | White | 18 | Renal (LN IV), arthritis, alopecia, pleuritis, hematologic | 3 | HCQ, AZA, MMF | HCQ, MMF, Pred 10 mg |

AZA azathioprine, BAR baricitinib, BEL belimumab, CsA cyclosporine A, CYC cyclophosphamide, HCQ hydroxychloroquine, IVIG intravenous immunoglobulins, LN lupus nephritis including the International Society of Nephrology/Renal Pathology Society (ISN/RPS) classification class (I-VI), MFA mycophenolic acid, MMF mycophenolate mofetil, MTX methotrexate, RTX rituximab. * Including continued immunosuppressants excluding glucocorticoids.

**Table 2 | Primary and secondary endpoints**

| | Baseline median (IQR) | Week 12 median (95% CI) | Change at Week 12 from Baseline median (95% CI) | Week 36 median (95% CI) | Change at Week 36 from Baseline median (95% CI) |
|---|---|---|---|---|---|
| **Primary endpoint** | | | | | |
| Anti-dsDNA antibodies (IU/ml) | 166.3 (87.3 – 351.0) | 61.1 (40.0 – 106.6) | −109.6 (−274.5 – −38.1) | 80.4 (27.8 – 127.2) | −123.0 (−236.4 – −34.2) |
| **Clinical secondary endpoints** | | | | | |
| SLEDAI-2K | 12 (10–14) | 4 (2 – 4) | -8 (-10 – -6) | 4 (2 – 10) | -9 (-10 – -2) |
| cSLEDAI-2K | 11 (8–12) | 2 (0 – 4) | -8 (-10 – -6) | 1 (0 – 2) | -10 (-14 – -10) |
| PGA (0-3) | 2.2 (2.1– 2.2) | 0.2 (0.1 – 0.3) | -2.0 (-2.1 – -1.8) | 0.3 (0.1 – 0.7) | -1.8 (-2.0 – -1.4) |
| SRI-4 response (%) | - | 100 (10/10) | | 70 (7/10) | |
| LLDAS rate (%) | - | 50 (5/10) | | 60 (6/10) | |
| Remission rate (%) | - | 20 (2/10) | | 50 (5/10) | |
| CLASI | 6.0 (3.0– 6.0) | 0.0 (0.0– 0.0) | -5.0 (-6.0 – -3.0) | 0.5 (0.0 – 3.0) | -4.0 (-6.0 – -1.0) |
| CDAI | 11.5 (4.0– 13.5) | 0.0 (0.0–1.0) | -11.0 (-13.5– -4.0) | 1.0 (0.0 – 4.0) | -9.5 (-11.0– -4.0) |
| UPCR (mg/g Creatinine)[1] | 649 (272– 1604) | 302 (117–768) | -269 (-690 – -84) | 325 (271 – 371) | -351 (-1261– -19) |
| Prednisolone dosage (mg/d) | 6.25 (5–9) | 6.25 (5–9) | 0 (0–0) | 5.0 (4–5) | -2.75 (-3.5–0) |
| SFI flare rate (%) | - | 0 (0/10) | | 20 (2/10) | |
| **Serologic secondary endpoints** | | | | | |
| C3 (mg/l) | 875 (690– 940) | 955 (810 – 1140) | 130 (80–230) | 1,045 (810 – 1,070) | 105 (-30 – 180) |
| C4 (mg/l) | 145 (100– 150) | 165 (130– 210) | 40 (30 – 60) | 125 (110 – 190) | 30 (-20 – 50) |
| IgG[2] (g/l) | 12.1 (8.9 – 19.9) | 6.9 (5.9 – 11.3) | −6.9 (−8.4 – −3.2) | 9.5 (6.0 – 14.1) | −3.0 (−4.4 – −2.0) |
| IgM[2] (g/l) | 0.6 (0.3– 0.9) | 0.2 (0.1– 0.4) | -0.2 (-0.5– -0.1) | 0.3 (0.2 – 0.7) | -0.1 (-0.2–0.0) |
| IgA[2] (g/l) | 2.3 (1.5– 2.9) | 0.6 (0.5– 0.8) | -1.5 (-2.0– -1.1) | 0.7 (0.6 – 1.3) | -1.0 (-1.6– −0.6) |
| Antinuclear antibody (reciprocal titres) | 640 (640– 5120) | 320 (320– 2560) | -320 (-320– -320) | 640 (160 – 2,560) | -400 (-480– -320) |
| Anti-SSA[3] (IU/ml) | 199.4 (78.3 – 201.9) | 134.3 (26.3– 185.7) | -20.3 (-67.6– -18.7) | 95.8 (46.4 – 188.8) | -16.9 (-70.3– -10.7) |
| Anti-RNP[4] (IU/ml) | 197.5 (59.5 – 354.5) | 169.4 (8.3–303.6) | -45.0 (-56.1– -18.9) | 190.7 (29.2– 316.8) | -31.0 (-50.2– -5.6) |
| Anti-Sm[5] (UI/ml) | 159.2 (72.8 – 245.6) | 96.3 (19.6–173.0) | −62.9 (-72.7– -53.2) | 110.7 (13.5–208.0) | -48.5 (-59.3– -37.6) |
| Anti-tetanus toxoid antibody levels (IU/ml)[2] | 1.7 (1.0 – 2.1) | 0.9 (0.4–1.5) | -0.6 (−1.0– −0.3) | 1.0 (0.6 – 1.9) | −0.1 (−0.5–0.2) |
| Anti-diphtheria antibody levels (IU/ml)[2] | 0.1 (0.09 – 0.5) | 0.2 (0.1–0.2) | -0.04 (-0.8 – -0.04) | 0.2 (0.1 – 0.4) | -0.04 (−0.4–(−0.03)) |
| **HR-QoL secondary endpoints** | | | | | |
| SF-36 (total) | 41.5 (35–54) | 74.5 (73–81) | 34.5 (11–46) | 71.5 (48–77) | 31 (0–36) |
| SF-36 (PCS) | 35.2 (31.2–47.6) | 55.0 (49.7–60.5) | 16.8 (6.0–19.3) | 53.5 (37.2–58.0) | 11.2 (0.2–19.1) |
| SF-36 (MCS) | 37.3 (30.7– 44.1) | 38.2 (33.1– 40.5) | -3.3 (-5.2–3.6) | 33.8 (32.3– 41.6) | -0.5 (-8.2–5.3) |
| FACIT-F | 19.5 (10.0–25.0) | 35.0 (26.0– 41.0) | 6.0 (3.0–23.0) | 38.5 (21.0– 41.0) | 8.5 (0.0–27.0) |

*IQR* Interquartile Range, *CI* confidence interval, *PGA* physician global assessment score, *UPCR* urine protein to creatinine ratio[1]. reported for 6 participants with active lupus nephritis at baseline[2], values are given for all participants, including all 5 participants that received a total of 9 doses of intravenous immunoglobulins[3]. reported for 5 participants with elevated anti-SSA antibodies at baseline[4]. reported for 6 patients with elevated anti-RNP antibodies at baseline[5] reported for 2 participants with elevated Anti-Sm antibodies at baseline.

occurred in three patients, all of which were with mild to moderate under treatment with nirmatrelvir/ritonavir. Further TEAEs included injection site reactions, fatigue and headache.

**Pharmacokinetics**

Given the presumably higher plasma cell burden in multiple myeloma compared to SLE, we hypothesized that the standard dosing regimen for daratumumab could result in supratherapeutic drug concentrations when applied to SLE. We analysed serum samples at week 9, i.e., one week after the last daratumumab injection, which is comparable to the time before cycle 3, day 1 of dosing in previous myeloma studies. Slightly higher trough concentrations ($C_{trough}$) were achieved with 1800 mg of weekly daratumumab in SLE than in the Phase 3 COLUMBA[30] and APOLLO[31] studies in multiple myeloma. The mean ± standard deviation (SD) daratumumab concentration in SLE was 1015 μg/ml (± 296 μg/ml), which was only comparable with the initial phase 1b PAVO trial[32] (Supplementary Fig. 2). Similar to data from the COLUMBA trial, daratumumab concentrations were increased in patients with lower body weight. Of note, the median body weight of our participants (65.5 kg, range 57–93) was lower than that of participants in the COLUMBA study (74.4 kg, range 39–103), which may partly explain the slightly higher $C_{trough}$ values observed in SLE.

**Treatment with daratumumab and dexamethasone reduces the number of NK cells and ASC, but not T cells**

To investigate the effects of the treatment on immune cells, we performed single-cell transcriptome (scRNA-seq) analysis and flow cytometry of peripheral blood mononuclear cells (Fig. 3A), including the use of a multi-epitope antibody that binds CD38 also in the presence of daratumumab. Total numbers of granulocytes, monocytes, dendritic cells, as wells as CD19⁺ B cells, CD4⁺ and CD8⁺ T cells remained largely stable, while the number of NK cells declined after the treatment

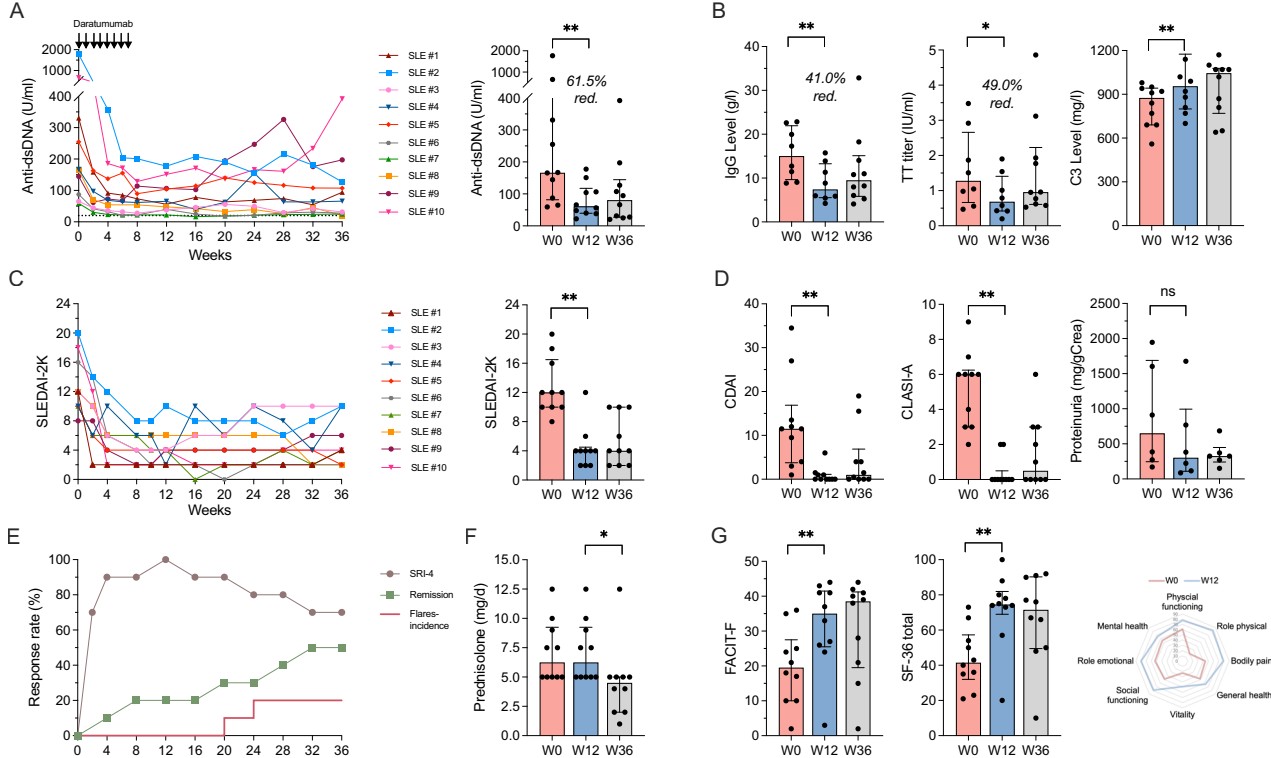

**Fig. 2 | Clinical and serologic effects of treatment with daratumumab and dexamethasone premedication.** Disease activity at baseline (W0), primary endpoint (W12) and last study visit (W36). **A** Changes in serum anti-double stranded (ds)DNA antibody levels (median change from baseline to W12 -109.6 IU/ml [95% CI -274.5 – -38.1, p = 0.002], to W36 -123.0 IU/ml [95% CI -236.4 – -34.2, p = 0.009]). **B** Immunoglobulin G (IgG) levels (median change from baseline to W12 -6.9 g/l [95% CI -8.4 – -3.2, p = 0.009], to W36 -3.0 g/l [95% CI -4.4 – -2.0, p = 0.106]) and anti-tetanus toxoid antibody (TT titres) (median change from baseline to W12 -0.6 IU/ml [95% CI -1.0 – -0.3, p = 0.009], to W36 -0.1 IU/ml [95% CI -0.5 – 0.2, p = 0.322]) and serum complement C3 levels (median change from baseline to W12 130 mg/l [95% CI 80 – 230, p = 0.002], to W36 105 mg/l [95% CI -30 – 180, p = 0.027]). Patients receiving IVIG before week 12 (n = 2) were excluded from the analysis. **C** SLEDAI-2K scores (median change from baseline to W12 -8 [95% CI -10 – -6, p = 0.002], to W36: -9 [95% CI -10 – -2, p = 0.004]) **D** Clinical Disease Activity Index (CDAI)

(median change from baseline to W12 -11.0 [95% CI -13.5 – -4.0, p = 0.004], to W36 -9.5 [95% CI -11.0 – -4.0, p = 0.006]), Cutaneous Lupus Disease Area and Severity Index (CLASI) (median change from baseline to W12 -5.0 [95% CI -6.0 – -3.0, p = 0.002], to W36 -4.0 [-6.0 – -1.0, p = 0.008]) and urinary proteinuria (in mg/g Creatinine). **E** Clinical response rates including SRI-4 and remission rates (as measured by the DORIS criteria) as well as cumulative flare incidence (as defined by the SLEDAI flare index (SFI)). **F** Daily prednisolone doses (median change from baseline to W12 0 mg/d [95% CI 0 – 0, p = 0.998], to W36 -2.75 mg/d [95% CI -3.5 – 0, p = 0.011]). **(G)** Scores of Functional Assessment of Chronic Illness Therapy (FACIT-F) (median change from baseline to W12 6.0 [95% CI 3.0 – 23.0, p = 0.002], to W36 8.5 [95% CI 0.0 – 27.0, p = 0.070]) and 36-Item Short Form Health Survey (SF-36) as self-reported fatigue and health-related quality of life scores. Median ± interquartile ranges are shown, a Wilcoxon signed-rank test was used to compared data and two-sided p values reported, n = 10 patients analysed unless indicated otherwise.

(median 0.08/nl at baseline, 0.01/nl at week 12; Supplementary Fig. 3A–D). NK cell counts recovered at week 24 with a similar distribution of the $CD16^+CD56^{dim/bright}$ phenotypes (Supplementary Fig. 3E). We found decreased levels of CD38-expressing $CD4^+$ and $CD8^+$ T cells ($CD8^+$ memory T cells: median 47,8% $CD38^+$ at baseline, 7.3% at week 12; Fig. 3B). Despite the strong reduction in $CD38^+$ T cells, T cell receptor (TCR) analysis of $CD8^+$ T cells demonstrated stable frequencies of highly abundant TCR clones after daratumumab (Fig. 3C, Supplementary Fig. 4). This suggests a persistence of formerly $CD38^+$ T cells that lose CD38 on the cell surface after daratumumab treatment by internalization or trogocytosis, similar to findings in multiple myeloma[33,34]. Within the $CD19^+$ B cell compartment, circulating ASC were preferentially depleted (median 1.5% ASCs among $CD19^+$ B cells at baseline, 0.61% at week 12; Fig. 2D, E). CD38 surface expression levels were reduced in dendritic cells, monocytes, NK cells and B cells (Supplementary Fig. 3D).

### Treatment with daratumumab and dexamethasone preferentially depletes mature $IgG^+$ ASCs

To better characterize the effects of the treatment on the ASC and B cell compartments, we performed scRNA-seq analyses. Peripheral

blood B and T cells were FACS-sorted at baseline, week 9 and week 36, yielding 137890 single-cell transcripts that were analysed and clustered into B cells, ASC, $CD4^+$ regulatory (Tregs), $CD4^+$ conventional T cells (Tcons) and $CD8^+$ T cells. (Supplementary Fig. 5A) Analysis of the ASC cluster identified a reduction of ASCs at week 9 after daratumumab, confirming the flow cytometry results (Supplementary Fig. 5B). Mature $CD138^+$ $IgG^+$ ASCs in the peripheral blood were recently reported to be associated with active SLE[4]. Transcriptional analysis of differentially expressed genes (DEGs) comparing ASC before and after daratumumab treatment showed a reduction of genes associated with a mature ASC status (*CD38, SDC1* (encoding CD138), *TNFRSF17* (encoding BCMA)) and antibody secretion (*XBP1, JCHAIN*) (Fig. 3F), suggesting a preferential depletion of mature ASCs, which was confirmed by a gene module score for ASC maturity (Supplementary Fig. 5C). In fact, ASC with a more mature plasma cell gene expression also had higher *CD38* (Supplementary Fig. 5D). In addition, single-cell B cell receptor sequencing demonstrated a preferential reduction of $IgG^+$ ASCs at week 9 after the start of treatment (Supplementary Fig. 5E).

The analysis of DEGs in B cells showed a reduction of genes that define age-associated B cells (ABCs), as well as type II polarised B cells

**Table 3 | Treatment-emergent adverse events (TEAEs) occurring during the study period**

| Event | Number of patients | Percent of patients | Number of events | Events per 10 years (95% confidence interval) |
|---|---|---|---|---|
| Any adverse event | **9** | **90** | **80** | **118.6 (94.5–146.5)** |
| Any severe adverse event | 0 | 0 | 0 | - |
| Hematologic | **5** | **50** | **10** | **14.8 (7.4–26.0)** |
| Immunoglobulin G < 5 g/L | 5 | 50 | 10 | 14.8 (7.4–26.0) |
| General | **6** | **60** | **18** | **26.7 (16.2–41.0)** |
| Injection site reaction | 3 | 30 | 12 | 17.8 (9.5–29.8) |
| Fatigue | 3 | 30 | 3 | 4.4 (1.1–11.5) |
| Influenca like illness | 2 | 20 | 2 | 3.0 (0.5–9.2) |
| Hot flush | 1 | 10 | 1 | 1.5 (0.1–6.5) |
| Infections | **8** | **80** | **27** | **40.0 (26.8–57.1)** |
| Nasopharyngitis | 5 | 50 | 8 | 11.9 (5.4–22.1) |
| COVID-19 | 3 | 30 | 5 | 7.4 (2.7–15.9) |
| Gastroenteritis | 3 | 30 | 3 | 4.4 (1.1–11.5) |
| Bronchitis | 2 | 20 | 3 | 4.4 (1.1–11.5) |
| Herpes zoster | 2 | 20 | 2 | 3.0 (0.5–9.2) |
| Bacteriuria | 1 | 10 | 1 | 1.5 (0.1–6.5) |
| Oral herpes | 1 | 10 | 1 | 1.5 (0.1–6.5) |
| Sinusitis | 1 | 10 | 1 | 1.5 (0.1–6.5) |
| Upper respiratory tract infection | 1 | 10 | 1 | 1.5 (0.1–6.5) |
| Urinary tract infection | 1 | 10 | 2 | 3.0 (0.5–9.2) |
| Gastrointestinal | **6** | **60** | **16** | **23.7 (13.9 - 37.3)** |
| Nausea | 4 | 40 | 8 | 11.9 (5.4–22.1) |
| Diarrhea | 3 | 30 | 6 | 8.9 (3.5–18.0) |
| Abdominal pain | 2 | 20 | 2 | 3.0 (0.5–9.2) |
| Musculoskeletal | **2** | **20** | **2** | **3.0 (0.5–9.2)** |
| Back pain | 2 | 20 | 2 | 3.0 (0.5–9.2) |
| Nervous System Disorders | **4** | **40** | **5** | **7.4 (2.7–15.9)** |
| Headache | 4 | 40 | 5 | 7.4 (2.7–15.9) |
| Respiratory | **1** | **10** | **1** | **1.5 (0.1–6.5)** |
| Dyspnea | 1 | 10 | 1 | 1.5 (0.1–6.5) |
| Skin and subcutaneous | **1** | **10** | **1** | **1.5 (0.1–6.5)** |
| Pityriaisis rosae | 1 | 10 | 1 | 1.5 (0.1–6.5) |

† total person years were 6.7 years

(Supplementary Fig. 5F–H)[35]. The respective clusters did not show changes in median abundance, partly due to the large interindividual differences in the B cell compartment and low *CD38* expression levels (Supplementary Fig. 5I–K). In addition, the expression of genes required for plasma cell differentiation (*FOS, SOX5*) was reduced in memory B cells after daratumumab and dexamethasone, consistent with a recent report showing reduced in vitro plasma cell differentiation in the presence of daratumumab[36]. Taken together, treatment with daratumumab and dexamethasone preferentially depletes circulating IgG+ ASC with a mature CD138+CD38high ASC status, potentially contributing to the beneficial clinical responses of the treatment.

**Daratumumab plus dexamethasone treatment modulates the functional profile of conventional and regulatory CD4+ T cells**

We next investigated the effect on peripheral blood CD4+ T cells, which remained unchanged in their number and memory subset distribution (central memory, effector memory and TEMRA cells) based on flow cytometry (Fig. 3G). We analysed single-cell transcriptomes of memory (i.e. not CD45RA+CCR7+) CD4+ conventional T cells (Tcons) that clustered along a differentiation gradient from central memory to effector memory. Unbiased clustering identified canonical subsets, including Th2 cells, Th17 cells and peripheral helper cells (Tph) (Supplementary Fig. 6A). None of these clusters changed in abundance after treatment. DEGs at baseline were dominated by type I IFN-induced genes (Fig. 3H) and a number of T cell-activation-associated transcripts, such as JUN family transcription factors (*JUN, JUND*), *NR4A2*, as well as *DUSP1* and *DUSP2*. While the type I IFN signature was reduced early after the treatment, transcripts associated with chronic activation were reduced later at the final study visit at week 36. Recently, increased CD38 expression on CD4+ T cells of SLE patients has been reported to induce dysfunction by increased store-operated calcium entry (SOCE) and subsequent endoplasmic reticulum (ER) stress[25]. Calculating an ER stress gene module score before and after treatment, we found a reduction at later time points, indicating that daratumumab-induced removal of CD38 may positively influence CD4+ T cell function (Supplementary Fig. 6B).

Next, we analysed transcriptomic changes in CD4+ regulatory T cells (Tregs), which suppress autoimmunity in healthy individuals and are dysfunctional in SLE[37–39]. Tregs showed little CD38 expression and had stable relative frequencies following treatment with daratumumab and dexamethasone, indicating no significant depletion (Supplementary Fig. 6C). This is in contrast to reports in multiple myeloma, where preferential targeting of CD38high Tregs was suggested to unleash a more efficient anti-tumour response[40]. Unbiased clustering reproduced the expected gradient of memory (expressing *CCR7, BACH2* and lower levels of *FOXP3*) and effector-type Tregs (expressing MHC-II molecules, *CTLA4*, and higher levels of *FOXP3*) (Supplementary Fig. 6D). While the relative cluster abundance did not change over time, the analysis of DEGs revealed a strong reduction of type I IFN-induced genes (like *MX1, IFI44L, XAF1, ISG15, IFIT1*) and an increase in gene expression indicating T cell receptor-mediated activation and function, including members of the AP-1 complex (*JUN, JUNB, JUND*), *FOXO1*, and *NR4A2* (Fig. 3I, Supplementary Fig. 6E). As chronic type I IFN signalling has been shown to induce Treg dysfunction[41,42], a reduced type I IFN signature and increased *FOXO1* expression which supports Treg stability[43] could potentially indicate improved Treg function.

**Daratumumab plus dexamethasone leads to normalization of CD38-induced CD8 T cell dysfunction**

In SLE, CD38high CD8+ cytotoxic lymphocytes (CTLs) have been reported to possess reduced cytotoxic capacity[24], due to epigenetic changes, mitochondrial dysfunction[26,28] as well as chronic type I IFN exposure[44]. This finding was associated with a clinically relevant increased risk of infection in SLE patients with high numbers of CD38high CTLs[45]. Similar to CD4 T+ cells, the distribution of CD8+ memory subsets was not changed after the treatment (Fig. 3J). Analyzing single cell transcriptomes, we identified major subsets of cytotoxic T cells, including *CCR7high* central memory, *GZMK+* effector memory, *GZMB+* effector cells and *KLRB1high* Mucosal-associated invariant T cells (MAITs), according to the expression of canonical markers. (Fig. 3K, Supplementary Fig. 6F). In agreement with our

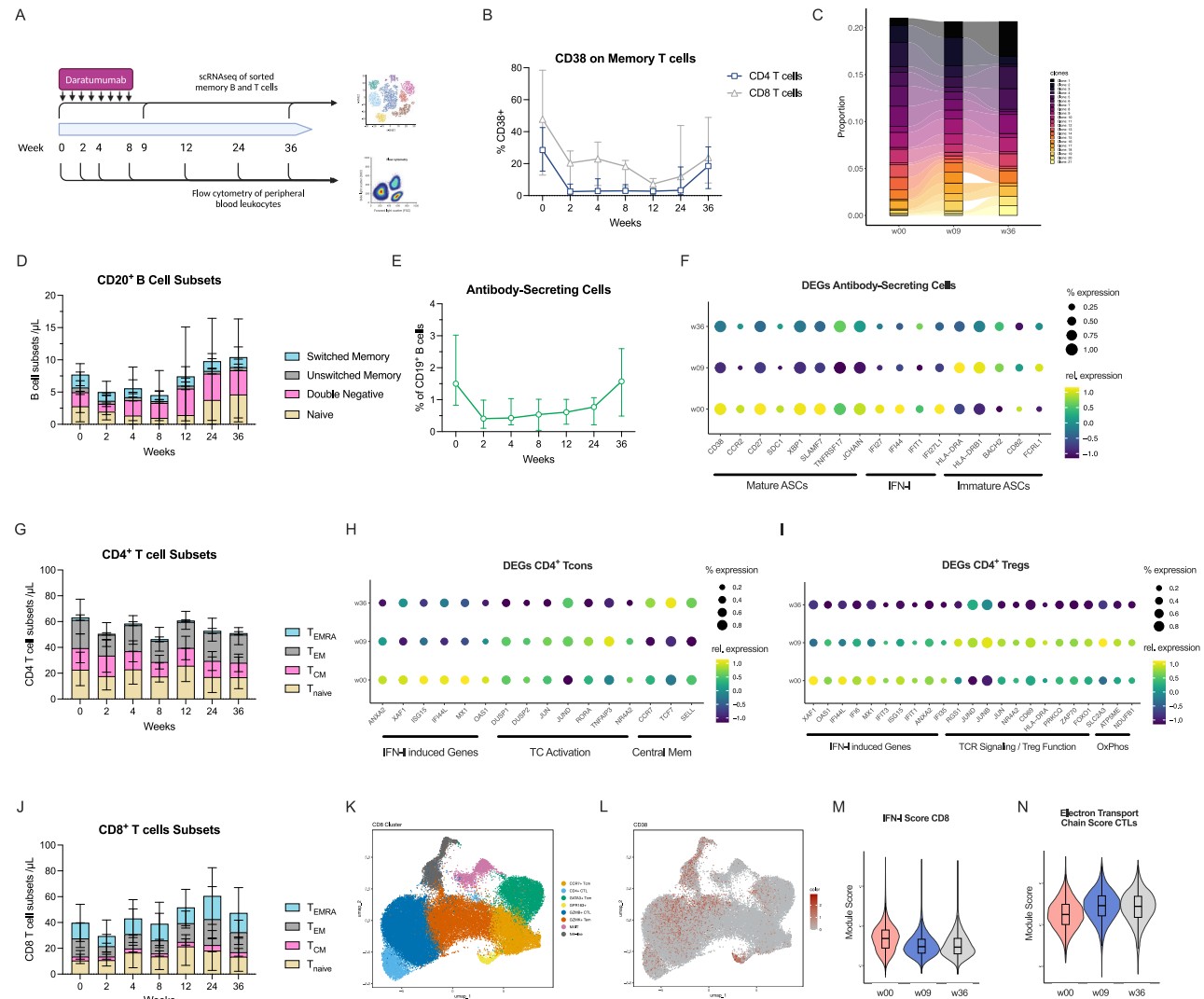

**Fig. 3 | Daratumumab with dexamethasone induces immunological changes in several immune cell types. A** Immunological investigations performed **B** Proportion of peripheral blood CD38-expressing CD4+ and CD8+ T cells. Median +/- interquartile ranges are shown. **C** The relative abundance of the most prominent CD8+ memory T cell clones at week 0, 9 and 36, as analysed by TCR sequencing, calculated as % of total TCR sequences. **D** Composition of the subsets of CD20+ B cells (non-ASC) defined as IgD−CD27+ switched memory B cells, IgD+CD27+ unswitched memory B cells, IgD−CD27− double negative B cells and IgD+CD27− naïve B cells as determined by flow cytometry. Median absolute counts +/- interquartile ranges are shown. **E** Relative counts of CD19low CD27high antibody-secreting cells (ASCs) among CD19+ B cells as determined by flow cytometry. Median and interquartile range are shown. **F** Expression of selected significant differentially expressed genes (DEGs) between timepoints in the ASC cluster. **G** Composition of the CD4+ memory T cells subsets, defined as CD45RA+CCR7− TEMRA (effector memory−expressing CD45RA), CD45RA−CCR7− TEM (effector memory), CD45RA−CCR7+ Tcm (central memory) and CD45RA+CCR7− Tnaive cells as determined by flow cytometry.

**H** Expression of selected significant DEGs between timepoints in the conventional CD4+ T cell cluster. **I** Expression of selected significant DEGs between timepoints in the regulatory CD4+ T cell cluster. **J** Composition of the CD8+ memory T cell subsets, defined as CD45RA+CCR7− TEMRA (effector memory−expressing CD45RA), CD45RA−CCR7− TEM (effector memory), CD45RA−CCR7+ Tcm (central memory) and CD45RA+CCR7+ Tnaive cells as determined by flow cytometry. **K** UMAP embedding of canonical clusters identified within single-cell transcriptomes of memory CD8+ T cells. **L** UMAP of the *CD38* expression within memory CD8+ T cells. **M** Violin plot of the hallmark "Interferon Alpha Response" gene module in CD8+ memory T cells. **N** Violin plot of the GOBP "Electron Transport Chain" gene module within the *GZMB*+ cytotoxic lymphocyte cluster. In barplots, median and interquartile ranges are shown. Boxplots indicates quartiles, whiskers indicate range. Flow cytometry includes n = 10 patients, scRNA from patients #6 and #8, as well as the week 36 time point of patient #4 are not available. Fig. 3A created in BioRender. Alexander, T. (2026) https://BioRender.com/rz99isu.

previous report[19], we identified the expression of CD38 in the *GZMB*+ subset of terminally differentiated effector CD8+ T cells. (Fig. 3L) Similar to CD4 T cells, DEGs comparing CTLs before and after treatment showed a strong reduction in type-I IFN-induced genes and a subsequent increase in genes associated with T cell activation (Fig. 3M, Supplementary Fig. 6G). Notably, there was a increase in genes encoding for components of the respiratory chain (such as *ATP5ME*), especially within the *GZMB*+ CTL cluster. (Fig. 3N) As CD38 induces mitochondrial dysfunction in CD8+ CTLs, these results suggest that

daratumumab treatment is associated with a reversal of type I IFN/CD38-induced metabolic and cytotoxic dysfunction in CD8+ T cells.

## Discussion

In this proof-of-concept phase 2 clinical trial, we investigated the safety and efficacy of the CD38-targeting monoclonal antibody daratumumab in patients with SLE who previously failed at least two immunosuppressive drugs. We found a significant depletion of circulating ASC and LLPCs, both of which are implicated in the disease

pathophysiology. Additionally, the treatment with daratumumab and dexamethasone as premedication resulted in a reduction of type I IFN activity and was associated with reduced dysfunction of CD8[+] T cells and regulatory T cells, which may have contributed to the observed beneficial clinical responses. The treatment was well-tolerated, and clinical responses occurred early, demonstrating efficacy across all major organ sites, collectively identifying daratumumab as a promising novel therapeutic approach, particularly as induction therapy for severely affected SLE patients.

The primary endpoint of the study – a reduction in serum anti-dsDNA antibodies at week 12—was met with a median dsDNA antibody reduction of 109.6 IU/ml. We chose this endpoint, as no previous data existed to select a realistic clinical endpoint, and because it provides mechanistic insight into the expected mode of action. These dsDNA antibodies are secreted by circulating and tissue-resident ASCs, both of which express high levels of CD38[5,22] and are therefore particularly susceptible to daratumumab-mediated depletion. In addition to autoantibody reduction, a significant decline in total IgG serum levels and vaccine-induced antibodies was evident, indicating a depletion of bone marrow-derived LLPCs. Additionally, flow cytometric and transcriptomic analyses confirmed a rapid depletion of ASCs in peripheral blood. Notably, ASC with a more mature IgG[+] plasma cell phenotype, as recently identified in active SLE[4], showed the highest *CD38* expression and were preferentially reduced by daratumumab treatment.

Reductions in serum autoantibodies were accompanied by rapid clinical improvement in all patients, reflected by a significant reduction in SLEDAI-2K scores and an SRI-4 response rate of 100% at week 12, respectively. Clinical efficacy included manifestations in all major organ domains, most prominently mucocutaneous and musculoskeletal manifestations. This is remarkable, considering that all patients had insufficient responses to at least two previous therapies, including B-cell-targeting therapies in 60% of the cases. This underscores the role of ASCs as potential drivers of autoimmune responses in SLE as previously indicated[11,12].

Clinically, daratumumab plus dexamethasone treatment resulted in a DORIS remission rate of 50% at week 36 which is consistent with significant clinically efficacy. Nevertheless, the treatment did not induce complete and treatment-free remission, as described in case reports of CD19 CAR-T cell therapy[46] or the BCMA-directed bispecific antibody teclistamab[15]. Similarly, anti-dsDNA antibodies were reduced by only about 61.5% at week 12, whereas these antibodies normalized under the aforementioned therapies. In addition, despite remarkable clinical improvements, recurrence of anti-dsDNA antibodies along with recurrence of circulating ASCs and disease activity was evident, with two patients experiencing flares at week 20 and 24, respectively. Thus, daratumumab does not appear to be a sufficient on-off therapy for SLE. Residual clinical disease activity was also present in non-flaring patients, indicating that the regimen of 8 doses of daratumumab is not sufficient to induce complete and sustained remission. It remains unclear whether extending daratumumab treatment as induction therapy would substantially improve clinical responses without disproportionately increasing the risk of adverse events. Alternatively, daratumumab could be effective in controlling residual or recurring disease activity when applied repeatedly as maintenance therapy. A recent case series provided evidence that daratumumab maintenance therapy induced stable responses even in the absence of background immunosuppressive treatments[20]. However, the dosing intervals required to control disease activity with the lowest risk of hypogammaglobulinemia and infections need to be determined. Alternatively, daratumumab could be combined with additional effective therapies to prevent regeneration of ASC, such as belimumab, which provided flare-free remissions over three years in two cases following daratumumab treatment[47].

Apart from the depletion of ASCs, we observed a strong decrease in CD38[+] memory T cells. The stable number of total memory T cells in

peripheral blood, along with the unchanged frequencies of clonally expanded CD8[+] T cells after treatment suggests that CD38 was removed from the surface of these cells, rather a depletion of these cells. This is presumably linked to the internalization or trogocytosis of CD38, similar to findings observed in multiple myeloma[33,34]. A number of recent reports have linked overexpression of CD38 to the dysfunction of CD4[+] and CD8[+] T cells, by inducing a number of epigenetic changes resulting in ER stress and accumulation of dysfunctional mitochondrial[24–28]. Using scRNA-seq, we demonstrated a general reduction of the type I interferon transcriptomic signature, a reduction in an ER stress gene signature in CD4[+] T cells and an increase in gene expression associated with mitochondrial respiration in CD8[+] T cells, likely indicating enhanced function of these cells after daratumumab treatment.

Despite the impaired humoral immunity associated with the depletion of LLPCs secreting protective antibodies and the strong reduction of NK cells early after treatment, we observed neither severe TEAEs nor TEAEs that led to discontinuation of the study. Most of the observed side effects included infections, gastrointestinal events, injection site reactions and hypogammaglobulinemia. IgG replacement therapy was initiated in five patients (50%), upon reaching serum IgG levels below 5 g/l, which may have reduced the incidence of infectious complications. IVIG substitution time points ranged from 4 weeks to 32 weeks following daratumumab, and a baseline IgG level below of 11.5 g/L appeared to predict hypogammaglobulinemia requiring IVIG substitution.

Our study has several limitations: First, the number of patients is relatively small for a phase 2 trial in SLE, particularly given the highly heterogeneous nature of this disease. The heterogeneity included the number of previous immunosuppressive drugs (ranging between 2 and 9 in addition to glucocorticoids) and the degree of renal disease (ranging from class IV/V combined lupus nephritis to no renal involvement). Nevertheless, the enrolled patients were homogeneous in that they all had high anti-dsDNA antibody levels, with mucocutaneous and musculoskeletal manifestations present in all patients. Second, treatment responses may be confounded by glucocorticoids given with each daratumumab injection. This procedure was selected according to the standard of care for administering of daratumumab in multiple myeloma[48]. Although glucocorticoids may be at least partly responsible for the rapid clinical responses, they are generally not associated with strong reductions in autoantibodies. Furthermore, all patients were clinically active despite continuous glucocorticoid use, with a median baseline dose of 6.25 mg prednisolone. During follow-up, the daily prednisolone dosage was significantly reduced, permitted after week 12, suggesting that long-term effects of glucocorticoids given as premedication are unlikely. The interpretation of the trial results is further complicated by the continued use of maintenance immunosuppressants and the use of belimumab as rescue therapy in two patients, making it more difficult to identify daratumumab-specific effects. Nevertheless, belimumab was started at week 20 and 24, respectively, thus not affecting the primary and secondary endpoints at week 12. We also performed a non-responder imputation of binary clinical endpoints after rescue treatment to reduce the confounding effect of rescue medication on long-term study outcomes. Additionally, the outcome measure Clinical Disease Activity Index (CDAI) is a tool to describe arthritis severity that is mainly used in rheumatoid arthritis and is not a validated measure in SLE. The CDAI was, however, chosen to assess musculoskeletal responses in more detail compared to the binary SLEDAI arthritis scoring. Finally, the follow-up period of 36 weeks is relatively short. To evaluate long-term safety and efficacy, we have included a long-term extension observational period of the DARALUP trial until week 84, which is currently ongoing.

In conclusion, our data demonstrate strong and rapid clinical improvement in all patients undergoing daratumumab treatment with

dexamethasone premedication, with a beneficial risk-benefit ratio. Treatment responses were primarily induced by a profound depletion of ASCs and LLPCs, resulting in significant reductions of anti-dsDNA antibodies. These data substantiate the central role of ASCs in the pathogenesis of SLE and identify daratumumab as a promising treatment option for SLE, particularly as induction therapy after insufficient responses to first-line immunosuppressive drugs. Additional mechanisms of action include immunomodulatory effects on T cells and the abrogation of type I interferon activity. These findings justify the further development of CD38-targeting antibodies in SLE, with controlled clinical trials needed to define their place in the treatment algorithm.

## Methods

### Trial design

The DARALUP study (Clinicaltrials.gov: NCT04810754; EudraCT number:2021-000962-14) is a single-arm, open-label, phase 2 study conducted at the Charité – Universitätsmedizin Berlin, Germany, in accordance with Good Clinical Practice guidelines of the International Council for Harmonisation and the principles of the Declaration of Helsinki. The study aimed to include a total of 10 patients. Sex and/or gender were not considered in the study design. The protocol (available as Supplementary Data 1) was approved by the institutional review board, and all the participants provided written, informed consent before beginning any trial-related procedures, including the publication of identifiable information.

### Participants

Eligible patients were 18 years of age or older who met the 2019 SLE classification criteria of the European Alliance of Associations for Rheumatology-American College of Rheumatology[49], demonstrated moderate to severe disease activity based on a SLEDAI-2K score ≥ 4 for clinical features despite conventional treatment (e.g. immunosuppressants, antimalarial drugs, corticosteroids) and failure of achieving remission or lack of tolerability with at least two prior disease modifying anti-rheumatic drugs. Furthermore, all participants were required to have increased serum anti-dsDNA antibody levels and stable background treatment using not more than 1 immunosuppressive drug (not including antimalarials and glucocorticoids). A washout period for previous treatments with rituximab or belimumab was required for 12 months and 3 months, respectively. Full eligibility criteria are provided in study protocol in the Supplementary Information in Supplementary Data 1. Patient deposition is detailed in the CONSORT chart (Fig. 1).

### Treatment plan

All participants continued their background medication at stable doses, except for oral glucocorticoids, which were allowed to be tapered after week 12. Daratumumab (at a dose of 1800 mg) was administered subcutaneously once a week for a total of 8 injections. Changes in dosing intervals were allowed if required, provided there was a minimum of 4 days between daratumumab doses. Premedication included 1000 mg of paracetamol and 50 mg of diphenhydramine orally, along with 20 mg of dexamethasone for the first two doses and 10 mg for all subsequent doses in all participants.

### Endpoints

The primary endpoint was the reduction in serum anti-dsDNA antibody titres at week 12, i.e., 4 weeks after the last daratumumab injection, compared to baseline. Anti-dsDNA antibodies were analysed by ELISA (Orgentec Diagnostica, Germany). Key secondary endpoints were safety and efficacy. Safety analyses were performed in the intention-to-treat population defined as all patients who received at least one dose of daratumumab. Patients were monitored for (S)TEAEs until week 36, and events were scored according to the National

Cancer Institute Common Terminology Criteria for Adverse Events (v.5.0)[50]. Efficacy assessment included analysis of the SLE responder index of 4 (SRI-4), DORIS remission rates[29], flare incidence according to the SELENA-SLEDAI flare index (SFI)[51], as well as health-related quality of life measures FACIT-F and SF-36 scores. Additional endpoints were immunologic changes and pharmacokinetics.

### Statistical analysis

The Statistical Analysis Plan is available in the Supplementary Data 2. All patients who were enrolled into the study were included in the main analyses. Baseline characteristics are presented for the intention-to-treat population defined as all patients enrolled in the study. Categorical variables are summarized as absolute numbers and percentages and continuous variables with medians, (interquartile) ranges and 95% CI intervals. The primary and secondary outcome measures were analysed according to the intention-to-treat principle. For the primary endpoint analysis, the median is reported for anti-dsDNA antibody titres at baseline and at week 12. The change between the two assessments was analysed using the non-parametric Wilcoxon signed-rank test and the median difference with 95% CI estimate were calculated. For secondary endpoints investigating continuous biomarker variables (except differential gene expression analysis; see below), differences between pre-, and posttreatment measurements were analysed using Wilcoxon's signed-rank test and the median difference with 95% CI estimate were calculated. Bootstrapping was performed in order to estimate the 95% CI of the median and median change in primary and secondary outcomes. The number of bootstrap replications was set to 1000, and bias-corrected confidence intervals were reported. The number of participants who achieved a SLE responder index of 4 (SRI-4) and clinical remission, respectively, are summarized graphically for each study visit. Disease flares were defined by the SELENA-SLEDAI flare index[51], the time to flare was analysed by Kaplan-Meier analyses. A non-responder imputation was performed for the two patients starting treatment with belimumab as rescue therapy at weeks 20 and 24 for the outcomes SRI-4, Remission and Flare incidence after treatment initiation of belimumab, respectively. All TEAEs were documented during daratumumab treatment. The incidence and the incidence rate per 10 treatment months of adverse events were calculated. All reported p values are two-sided, and a p value of <0.05 was considered statistically significant. Analyses were performed using R v.4.2.2.

For all study visits, the rate of missing data was <0.5% for all primary and secondary outcomes. The outcome data for the predefined week 0, week 12 and week 36 time points of analysis was complete, therefore no imputation of missing data was performed.

For the immunological data, the Chi-Squared test, Kruskal-Wallis test with Dunn's test for multiple comparisons or Friedman test with Dunn's test for multiple comparisons were computed as appropriate in R v.4.4.0 or GraphPad Prism v10.4.1 for MacOS and indicated in the figure legend. Two-sided p values were reported.

### Flow cytometry

Peripheral blood mononuclear cells (PBMCs) were isolated from heparin-anticoagulated blood at week 0, 2, 4, 8, 12, 24, and 36 using Ficoll-Paque PLUS gradient (GE Healthcare) and resuspended in freeze medium (90% fetal calf serum/10% DMSO) and stored at -80 °C until cytometry. For flow cytometry, the samples were thawed in a 37 °C water batch and quickly transferred in RPMI 1640 medium with Glutamax supplement (ThermoFisher). After washing, samples were stained with fluorophore-coupled antibodies (Supplementary Table 3) and measured on a BD Fortessa™ cytometer (BD Biosciences). Due to insufficient viability, week 24 samples of patients #7 and #10 were excluded. To avoid batch effects, all samples from a single patient were analysed on the same day. The flow cytometry data were analysed using FlowJo 10.10.0 for MacOS (FlowJo). Example gating from the data analysis is shown in Supplementary Fig. 7. Flow cytometry and cell

sorting was performed in accordance with published best practice guidelines[52].

## Single cell transcriptome and immune cell receptor sequencing

At week 0, 9, and 36, PBMCs were isolated from fresh heparin-anticoagulated blood using Ficoll-Paque PLUS gradient (GE Healthcare), washed, and incubated with fluorescent antibodies against CD3, CD19, CD45RA, CCR7, and CD27. In case more than one patient sample was analysed at the same day, TotalSeq™-C anti-human Hashtag antibodies (BioLegend) were used. Using a BD AriaII™ cell sorter, we sorted both CD19⁺CD27⁺ memory B cells/plasmablasts and CD3⁺ non-naïve (i.e., not positive for both CCR7 and CD45RA) T cells, and adjusted cell numbers to sequence 10.000 cells per patient and time point.

Single-cell RNA library construction and sequencing was done as previously described[5,53]. Briefly, Chromium Next GEM Single Cell 5' reagent kits v2 (dual index) with feature barcode technology for cell surface protein (CITE) mapping (10X Genomics) were used according to the manufacturer's protocol. Final CITE-Seq libraries were generated after index PCR with dual Index Kit TN Set A (10X Genomics) while final GEX and TCR/BCR libraries were generated after fragmentation, adapter ligation and final index PCR with a dual Index Kit TT Set A (10X Genomics). Libraries were quantified using a Qubit HS DNA assay kit (Life Technologies) and fragment sizes were determined using a HS NGS Fragment (1-6000 bp) kit (Agilent). All libraries were sequenced on a NextSeq2000 sequencer (Illumina). GEX and CITE libraries were sequenced using a P3 reagent cartridge (100 cycles) (Illumina) with the following recommended sequencing conditions: read1: 26 nt, index1: 10 nt, read2: 90 nt, index2: 10 nt. The BCR and TCR libraries spiked-in with 10% 2 nM PhiX were sequenced using a P3 reagent cartridge (300 cycles) (Illumina) with the following recommended sequencing conditions: read1: 151 nt, index1: 10 nt, read2: 151 nt, index2: 10 nt. Due to organisational/technical problems, data from patients #6 and #8, as well as the week 36 time point of patient #4 had to be excluded.

## Single cell transcriptome data analysis and immune profiling

Raw sequence reads were processed using cellranger (version 7.1.0). Demultiplexing, mapping, detection of intact cells, as well as quantification of gene expression, were performed using cellranger's count pipeline in default parameter settings with refdata-gex-GRCh38-2020-A as reference, Hashtag 1-10 as feature reference, and an expected number of 3000 cells per sample. Cellranger's aggr function was used to merge the libraries without size and reanalyse without Immunoglobulin and TCR genes which otherwise produce artefacts. The raw sequence reads from immune profiling were also processed using cellranger (version 7.1.0), where cellranger's vdj pipeline was used in default parameter settings for demultiplexing and assembly of the TCR and BCR sequences using refdata-cellranger-vdj-GRCh38-alts-ensembl-2.0.0 as reference.

The resulting data were analysed in R[54] (v 4.4.1) using the standard Seurat[55] workflow (v.5.1.0). Transcriptome profiles were imported using Read10x and CreateSeuratObject functions and data normalization, identification of variable features and scaling were done with default settings. Afterwards integration of the samples from different batchIDs was done using harmony[56] (v.1.2.1) to remove batch effects. Based on 20 dimensions of the harmony reductions, FindNeighbours and RunUMAP were run. Samples comprising two patients or donors were demultiplexed by gating of hashtag counts in a scatterplot after arcsinh transformation. Annotations of the TCR and BCR were assigned to the corresponding cells in the single-cell transcriptome analysis by identical cellular barcodes. In case of multiple contigs, the most abundant, productive, and fully sequenced contig for the alpha and beta chain for TCR and heavy and light chain for BCR, respectively, was used. T cell receptor and B cell receptor repertoirs were analyzed using the scRepertoire package (v3.20)[57]. The overlap of the most prominent TCRs (global, top 10 of each time point) was assessed with the clonalCompare function in scRepertoire.

Shared nearest neighbor (SNN)-based clustering was performed based on the harmony reduction and clusters corresponding to CD4+ conventional T cells, CD4+ regulatory T cells, CD8 + T cells, proliferating cells, Memory B cells and plasmablasts were identified based on the expression of canonical markers (*CD3E, CD4, CD8A, FOXP3, IKZF2, MKI67, MS4A1, CD79B, CD38, JCHAIN, SDC1*). Consequently, each cell type was sub-clustered, and re-integration, clustering, and UMAP calculation were done with the same routine as for total cells. Differential gene expression between the timepoints was assessed using the FindAllMarker function. In the standard settings we used, first genes that have a log2 fold change difference in expression of at least 0.1 are filtered and a Wilcoxon Rank Sum test is used to calculate p values with bonferroni correction based on the total number of genes in the dataset. The top differentially expressed genes were selected based on fold changes and absolute difference in expression and functional relevance based on the literature. All gene expressions shown in the dot plots were significantly differentially expressed with adjusted p values < 0.05. The gene sets "HALLMARK_INTERFERON_ALPHA_RESPONSE", "GOBP_ELECTRON_TRANSPORT_CHAIN" and "GOBP_RESPONSE_TO_ENDOPLASMIC_RETICULUM_STRESS" in their most current version as of November 2024 were used to assess module scores using the AddModuleScore function. Similarly, a plasma cell score was calculated using the gene expression of *CD38, CCR2, CD27, SDC1, XBP1, SLAMF7, TNFRSF17, JCHAIN* and *PRDM1*. Feature plots, dot plots and violin plots were generated using the dittoSeq library (v1.16.0)[58]. The overview of immunological investigations (Fig. 3A) was produced using biorender.com.

## Reporting summary

Further information on research design is available in the Nature Portfolio Reporting Summary linked to this article.

# Data availability

Patient-identifiable data cannot be shared. Source data for the figures are deposited with this paper. Further non-identifiable data can be obtained from the corresponding author upon request, subject to specific criteria, including the nature of the research inquiry and the required ethical approvals. Single cell transcriptome sequencing data is available at GEO under the accession code GSE294789. [https://www.ncbi.nlm.nih.gov/geo/query/acc.cgi?acc=GSE294789] Source data are provided with this paper.

# Code availability

All analyses were performed using publicly available published tools and resources, as described in the methods section. No custom code was generated.

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

## Acknowledgements

This study was supported by Janssen-Cilag GmbH, Johnson & Johnson, which also provided the investigational drug for this study. L.O. was supported by the BIH Junior Charité Clinician Scientist Program funded by the Charité – Universitätsmedizin Berlin, and the Berlin Institute of Health at Charité (BIH), and received support from the European Union as part of the Marie Skłodowska-Curie Action fellowship (Project 101153683) and under grant agreement no. 101057438 (geneTIGA). Views and opinions expressed are those of the author(s) only and do not necessarily reflect those of the European Union or the European Health and Digital Executive Agency (HADEA). Neither the European Union nor the granting authority can be held responsible for them. This work was further supported by the Federal Ministry of Education and Research (BMBF), providing support with financing of the projects TReAT and CONAN to M.-F.M.; the state of Berlin and the European Regional Development Fund through the grant EFRE 1.8/11 and EFRE PersMedLab to M.-F.M.; the Leibniz Association through the Leibniz Collaborative Excellence TargArt and the ImpACt project to M.-F.M. the German Center for Child and Adolescent Health (DZKJ) grants 01GL2401C to M.-F.M., the state of Berlin through 'Systematic decoding of unclear inflammations in children by single-cell sequencing and artificial intelligence". We thank Skander Elleuche from EUROIMMUNE AG for the assessment of extractable antinuclear antibodies (ENA).

## Author contributions

L.O., J.K., F.H., M.F.M., and T.A. planned the study. L.O., J.Z., R.K., A.E.B., R.B., Q.C., L.K., G.M.G., G.R.B., G.K., F.H., and T.A. contributed to data acquisition. L.O., J.K., F.H., P.D., M.F.M., and T.A. contributed to data analysis, M.F.M. and T.A. supervised the work, G.K., F.H., and T.A. acquired funding, L.O. and T.A. wrote the original manuscript, and all authors gave important input to the original manuscript and its revision.

## Funding

## Competing interests

RB received honoraria from AstraZeneca and G.S.K. F.H. received honoraria from AstraZeneca. T.A. received study support from Janssen-Cilag GmbH and honoraria from Amgen, AstraZeneca and G.S.K. The remaining authors declare no competing interests.
