## [Transparent Peer Review file · Nature Communications]

Daratumumab in systemic lupus erythematosus: a single-arm phase 2 trial

Corresponding Author: Dr Tobias Alexander

Version 0:

Reviewer comments:

Reviewer #1

(Remarks to the Author)

I have reviewed with great interest this phase 2, single-arm, open-label clinical trial evaluating the safety and efficacy of daratumumab—an anti-CD38 monoclonal antibody that depletes antibody-secreting cells (ASCs)—in 10 women with active systemic lupus erythematosus (SLE) who had inadequate responses to at least two immunosuppressive therapies.

From my understanding, enrolled participants received eight weekly subcutaneous doses of daratumumab (1800 mg) with dexamethasone premedication. By week 12, serum anti-double-stranded DNA antibody levels were significantly reduced by a median of 61.5%, accompanied by clinical improvement in all patients, reflected in a 100% SRI-4 response rate. Treatment also reduced circulating ASCs, type I interferon activity, and modulated T-cell responses. Hypogammaglobulinemia occurred in half of the patients, requiring immunoglobulin replacement. The findings support daratumumab as a promising therapeutic option targeting ASCs in SLE.

- It was difficult to comprehend whether all of these "story outlines" were directly reflected by the outcomes and timepoints registered with clinical trials.gov (ie, key secondary endpoints).

Q1: While reporting frameworks for single-arm trials are not rigorously defined, I find it problematic - speaking as a biostatistician in clinical epidemiology - that the authors attempt to impress readers in their narrative by reporting P-values. As the authors are likely already aware, P-values beyond a clearly defined primary endpoint tied to a specific objective are highly susceptible to spurious findings and are, at best, difficult to interpret.

- Please, could the authors make sure to report all their estimates (ie, most likely median changes from baseline) with the appropriate 95% confidence intervals

Q2: It is critical that the authors make it absolutely clear – in the abstract in particular - that the following outcome reflected the primary objective

“Change in serum anti-dsDNA antibody titers” [Time Frame: Week 12 (i.e. 4 weeks after last daratumumab injection)]

- If the authors insist that the percentage change is more important than the actual change; then, please make sure to add that to the abstract too.

- In the abstract reporting, please omit the P-value, and simply imply that there was on average a reduction in autoantibodies; please support the median change with the anticipated 95% confidence intervals.

Q2: Secondary Outcomes.

Please, will the authors confirm that they report all of the following outcomes, listed as endpoints - in the trial registration - in their main outcome table (Table 2):

Secondary (Current) [*]ICMJE (Submitted: 2021-03-19)

- Assess the Incidence of Treatment-Emergent Adverse Events [Time Frame: through study completion, from screening up to Week 36]

Adverse Events will be as assessed and graded by Common Terminology Criteria for Adverse Events (CTCAE) v5.0.

- Clinical outcome parameter [Time Frame: Week 12]

Number of participants achieving Systemic Lupus Responder Index 4

- SLE serology [Time Frame: through study completion, up to Week 36]
Evaluating the change in serum complement factor C3 levels
- GC sparing [Time Frame: between Week 12 and Week 36]
Investigating median change of daily prednisolone dosage
- Health-related quality of life [Time Frame: through study completion, up to Week 36]
Patient related outcome measures will be investigated, overall health assessed by Short-Form 36 Score
REMEMBER THESE MEASURES ARE AT LEAST 2: PCS and MCS, respectively
Health-related quality of life [Time Frame: through study completion, up to Week 36]

- Patient related outcome measures will be investigated, Fatigue assessed by Functional Assessment of Chronic Illness Therapy - Fatigue (FACIT-F) score

Q3: If the authors feel strongly about reporting their primary endpoint as a percentage change from baseline, please ensure that the corresponding statistical estimates are also included in Table 2.

Q4: Regarding Figure 2 and its legend: If the authors believe that all the reported significant P-values are important, then at a minimum, please also include the corresponding 95% confidence intervals. This will provide a clearer sense of the precision of the estimates and help clinicians judge the clinical relevance of the findings.

Q5: Figure 3. Please remove all P-values from the figure legend. If the authors believe their findings are important, they should state so directly—P-values do not offer meaningful support in this context.

Reviewer #2

(Remarks to the Author)

Ostendorf and colleagues report on the use of daratumumab in systemic lupus erythematosus, providing a well-conducted study that yields important translational insights. The work is particularly strong in its detailed characterization of the immunological effects of anti-CD38 therapy, though it reads more as an immunological exploration than a definitive clinical efficacy and safety trial. The findings on immune modulation are compelling and advance our understanding of SLE pathophysiology. Nevertheless, clinical interpretation is tempered by factors such as cohort heterogeneity, variability in prior treatments, and differences in baseline immunological activity. Moreover, the administration of daratumumab as add-on therapy limits assessment of its stand-alone clinical value.

Overall, while the study provides valuable translational insights, several elements of the design and patient population merit further consideration and discussion.

1) the inclusion of background immunosuppressive therapies (e.g., azathioprine, mycophenolate mofetil/mycophenolic acid) and the use of rescue agents (e.g., belimumab) should be more clearly contextualized within the framework of a Phase II trial. While clinically justified, their concomitant use limits assessment of the standalone safety and efficacy of daratumumab.

2) The study population is relatively heterogeneous. Nine of ten patients had a history of lupus nephritis; however, one lacked a kidney biopsy and two had only Class II disease.

3) Two patients had received only two or three disease-modifying antirheumatic drugs, including hydroxychloroquine combined with azathioprine and/or mycophenolate mofetil. This raises the question of whether these individuals represent the most suitable target group for evaluating anti-CD38 therapy, as more treatment-refractory patients might better elucidate its potential benefit.

4) The optimal daratumumab regimen for maintaining remission remains uncertain. Two patients (patients #3 and #4) experienced flares at weeks 20 and 24 requiring belimumab, and anti-dsDNA antibody levels increased in six patients during follow-up. At study end, SLEDAI-2K scores remained at 4, indicating residual disease activity. Six patients continued on background immunosuppression (azathioprine or mycophenolate mofetil), making it difficult to determine whether daratumumab alone could sustain remission.

Reviewer #4

(Remarks to the Author)

This manuscript describes the immunologic effects of treatment with daratumumab, along with a high dose of corticosteroid premedication, in patients with high levels of lupus activity and positive anti-dsDNA despite prior treatment with multiple immunosuppressive medications. My comments are essentially restricted to a review of the assessment by former reviewer #3 and the authors' responses to those suggestions.

In general, I am comfortable with the modifications and explanations of the authors. My remaining concerns are the following:

1) The very high dose of dexamethasone premedication remains a concern, especially if a controlled phase 2 study is to be

designed based on these results. How did the investigators choose this dose? Is there data to support it?

2) Table 1 provides information regarding the daily prednisone dose during the initial phase of the drug trial. No data is provided regarding prior glucocorticoid doses in these patients, such as peak or cumulative dose. Had the study participants received large steroid doses during prior unsuccessful treatment, this might mitigate the concern that the study steroid dose was actually responsible for the therapeutic benefit.

3) Given the concerns regarding dexamethasone premedication, the dose of dexamethasone should be added to line 44 in the abstract.

4) Reviewer #3's concern in point #2 is relevant, and the authors' response is appropriate and helpful. I suggest adding both the question and the response to the discussion section of the manuscript.

Version 1:

Reviewer comments:

Reviewer #1

(Remarks to the Author)

Many thanks for the constructive clarifications. They reflect a careful and transparent approach to reporting this important single-arm phase 2 trial.

My methodological suggestions were not criticisms but opportunities to strengthen clarity, transparency, and reproducibility - especially given the challenges of interpreting small single-arm studies.

I am confident that the modest additions will help the manuscript fully reflect the rigor of your work.

I am happy to confirm that I fully concur with the latest version.

Reviewer #2

(Remarks to the Author)

comments were addressed and the manuscript improved in clarity through the review process.

Reviewer #4

(Remarks to the Author)

I am satisfied that the authors have revised the manuscript in accordance with my suggestions, relevant to the recommendations of the original reviewer #3.

Dear Editors,

we are grateful for the opportunity to submit a revised manuscript for publication. We uploaded the manuscript with changes tracked according to the reviewer's comments:

Reviewer #1 (Remarks to the Author):

From my understanding, enrolled participants received eight weekly subcutaneous doses of daratumumab (1800 mg) with dexamethasone premedication. By week 12, serum anti-double-stranded DNA antibody levels were significantly reduced by a median of 61.5%, accompanied by clinical improvement in all patients, reflected in a 100% SRI-4 response rate. Treatment also reduced circulating ASCs, type I interferon activity, and modulated T-cell responses. Hypogammaglobulinemia occurred in half of the patients, requiring immunoglobulin replacement. The findings support daratumumab as a promising therapeutic option targeting ASCs in SLE.

- It was difficult to comprehend whether all of these "story outlines" were directly reflected by the outcomes and timepoints registered with clinical trials.gov (ie, key secondary endpoints).

Q1: While reporting frameworks for single-arm trials are not rigorously defined, I find it problematic - speaking as a biostatistician in clinical epidemiology - that the authors attempt to impress readers in their narrative by reporting P-values. As the authors are likely already aware, P-values beyond a clearly defined primary endpoint tied to a specific objective are highly susceptible to spurious findings and are, at best, difficult to interpret.

- Please, could the authors make sure to report all their estimates (ie, most likely median changes from baseline) with the appropriate 95% confidence intervals

We are grateful for your remarks. According to your suggestion, we have adapted the legend of Figure 2 to include estimates of median changes and report median changes and corresponding 95% CI. Likewise, we included the absolute changes of anti-dsDNA antibodies alongside the 95% CI in the abstract.

Q2: It is critical that the authors make it absolutely clear – in the abstract in particular - that the following outcome reflected the primary objective

“Change in serum anti-dsDNA antibody titers” [Time Frame: Week 12 (i.e. 4 weeks after last daratumumab injection)]

- If the authors insists that the percentage change is more important than the actual change; then, please make sure to add that to the abstract too.

- In the abstract reporting, please omit the P-value, and simply imply that there was on average a reduction in autoantibodies; please support the median change with the anticipated 95% confidence intervals.

We now clarified in the abstract the primary objective, i.e. reduction in serum anti-double-stranded DNA (anti-dsDNA) antibody levels at week 12. We also deleted the p-value in the abstract according to your suggestions and refocused the reporting of the outcome to absolute

reductions in autoantibodies.

Q2: Secondary Outcomes.

Please, will the authors confirm that they report all of the following outcomes, listed as endpoints - in the trial registration - in their main outcome table (Table 2):

Secondary (Current) [*]ICMJE (Submitted: 2021-03-19)

- Assess the Incidence of Treatment-Emergent Adverse Events [Time Frame: through study completion, from screening up to Week 36]

Adverse Events will be as assessed and graded by Common Terminology Criteria for Adverse Events (CTCAE) v5.0.

- Clinical outcome parameter [Time Frame: Week 12]

Number of participants achieving Systemic Lupus Responder Index 4

- SLE serology [Time Frame: through study completion, up to Week 36]

Evaluating the change in serum complement factor C3 levels

- GC sparing [Time Frame: between Week 12 and Week 36]

Investigating median change of daily prednisolone dosage

- Health-related quality of life [Time Frame: through study completion, up to Week 36]

Patient related outcome measures will be investigated, overall health assessed by Short-Form 36 Score

REMEMBER THESE MEASURES ARE AT LEAST 2: PCS and MCS, respectively

Health-related quality of life [Time Frame: through study completion, up to Week 36]

- Patient related outcome measures will be investigated, Fatigue assessed by Functional Assessment of Chronic Illness Therapy - Fatigue (FACIT-F) score

We double-checked and can confirm that all the secondary endpoints listed in the trial registration are reported in the main outcome table 2, except for the adverse events, which are reported in the separate adverse events table 3.

Q3: If the authors feel strongly about reporting their primary endpoint as a percentage change from baseline, please ensure that the corresponding statistical estimates are also included in Table 2.

We thank the reviewer for this comment, we have changed the focus of the reporting of the primary endpoint to the absolute change from baseline.

Q4: Regarding Figure 2 and its legend: If the authors believe that all the reported significant P-values are important, then at a minimum, please also include the corresponding 95% confidence intervals. This will provide a clearer sense of the precision of the estimates and help clinicians judge the clinical relevance of the findings.

Thanks for your suggestion. We have added the 95% confidence intervals to the figure 2 and its legend.

Q5: Figure 3. Please remove all P-values from the figure legend. If the authors believe their findings are important, they should state so directly—P-values do not offer meaningful support in this context.

We have removed the P-values from Figure 3 and also the corresponding supplemental figures.

Reviewer #2 (Remarks to the Author):

Ostendorf and colleagues report on the use of daratumumab in systemic lupus erythematosus, providing a well-conducted study that yields important translational insights. The work is particularly strong in its detailed characterization of the immunological effects of anti-CD38 therapy, though it reads more as an immunological exploration than a definitive clinical efficacy and safety trial. The findings on immune modulation are compelling and advance our understanding of SLE pathophysiology. Nevertheless, clinical interpretation is tempered by factors such as cohort heterogeneity, variability in prior treatments, and differences in baseline immunological activity. Moreover, the administration of daratumumab as add-on therapy limits assessment of its stand-alone clinical value.

Overall, while the study provides valuable translational insights, several elements of the design and patient population merit further consideration and discussion.

1) the inclusion of background immunosuppressive therapies (e.g., azathioprine, mycophenolate mofetil/mycophenolic acid) and the use of rescue agents (e.g., belimumab) should be more clearly contextualized within the framework of a Phase II trial. While clinically justified, their concomitant use limits assessment of the standalone safety and efficacy of daratumumab.

We agree with the reviewer that background immunosuppressive therapies are an important aspect in trial design in SLE and lupus nephritis. In fact, many of the recent large trials of novel treatments in SLE included the continuation of background immunosuppressive treatments — for example the recent anifrolumab trial in SLE (DOI: 10.1056/NEJMoa1912196). This approach is required for ethical reasons because of the uncertainty regarding the efficacy of the study drug. Similarly, it is unethical and unfeasible to have patients with new flares of SLE disease activity within a clinical trial and a standardized per-protocol rescue treatment approach reduces heterogeneity in flare treatment in the study population. To exclude the confounding effects on rescue medication on the study outcome, we have performed a non-responder imputation for the binary clinical outcomes like SRI-4 response and remission to not skew these results by rescue treatment. We clarified this issue in the discussion part of the manuscript.

2) The study population is relatively heterogeneous. Nine of ten patients had a history of lupus nephritis; however, one lacked a kidney biopsy and two had only Class II disease.

We agree with the reviewer on this point. The heterogeneity of clinical features is certainly related to the relatively low number of included individuals. Nevertheless, the study population is homogenous in the primary scope of this pilot study, i.e. persistent serologic AND clinical activity despite standard treatment. Although LN was not the main focus of this study, we have added the nephritis class information in the main text and also mentioned this aspect in the discussion section.

3) Two patients had received only two or three disease-modifying antirheumatic drugs, including hydroxychloroquine combined with azathioprine and/or mycophenolate mofetil. This raises the question of whether these individuals represent the most suitable target group for evaluating anti-CD38 therapy, as more treatment-refractory patients might better elucidate its potential benefit.

We fully agree with the reviewer's comment – the ideal target population for an anti-CD38 therapy is currently unknown and the current study results are not sufficient to address this question. We could speculate that either more refractory patients with more established pathogenic plasma cell memory benefit more or that a plasma cell-targeted therapy early in the disease course could stop the accrual of damage in the disease course. In our small trial, we have not seen a clear signal correlating “refractoriness” and response to anti-CD38 treatment. Further controlled trials will now be needed to identify the groups of SLE patients that can benefit from anti-CD38 monoclonals.

4) The optimal daratumumab regimen for maintaining remission remains uncertain. Two patients (patients #3 and #4) experienced flares at weeks 20 and 24 requiring belimumab, and anti-dsDNA antibody levels increased in six patients during follow-up. At study end, SLEDAI-2K scores remained at 4, indicating residual disease activity. Six patients continued on background immunosuppression (azathioprine or mycophenolate mofetil), making it difficult to determine whether daratumumab alone could sustain remission.

We agree with the reviewer, the role of the ideal daratumumab regimen in general, and for maintenance therapy specifically is unknown and cannot be answered from this trial. It remains unclear whether increasing the number of daratumumab injections for induction would lead to substantially improved clinical efficacy without increasing the risk of infections. Our data indicate a plateau effect of daratumumab on clinical, and more importantly serologic changes, where most of the reduction in autoantibodies occurs with the first doses and additional doses showed only marginal further improvement. Conceptionally, daratumumab could be continued as maintenance therapy, or combined with additional B-cell targeting agent like belimumab or obinutuzumab to prevent regeneration of plasma cells. Future trials will be needed to investigate these important research questions. We extended the discussion part of the manuscript accordingly.

Reviewer #4 (Remarks to the Author):

This manuscript describes the immunologic effects of treatment with daratumumab, along with a high dose of corticosteroid premedication, in patients with high levels of lupus activity and positive anti-dsDNA despite prior treatment with multiple immunosuppressive medications. My comments are essentially restricted to a review of the assessment by former reviewer #3 and the authors' responses to those suggestions. In general, I am comfortable with the modifications and explanations of the authors. My remaining concerns are the following:

1) The very high dose of dexamethasone premedication remains a concern, especially if a controlled phase 2 study is to be designed based on these results. How did the investigators choose this dose? Is there data to support it?

The dexamethasone dose used was chosen based on the established premedication protocol recommended by the manufacturer and follows the standard-of-care for the use of daratumumab in multiple myeloma. Although confounding the results of daratumumab treatment in SLE, we kept this standard protocol for safety reasons.

2) Table 1 provides information regarding the daily prednisone dose during the initial phase of the drug trial. No data is provided regarding prior glucocorticoid doses in these patients, such as peak or cumulative dose. Had the study participants received large steroid doses during prior unsuccessful treatment, this might mitigate the concern that the study steroid dose was actually responsible for the therapeutic benefit.

We fully agree that the effect of previous steroid-treatment may help to put the dexamethasone given as premedication into perspective. Accordingly, we added that the continuous use of steroids was required in all patients. All patients were refractory to previous immunosuppressants, including glucocorticoids, and most had been treated for years prior to the study, including in other centers. We were therefore not able to ascertain peak doses and cumulative doses.

3) Given the concerns regarding dexamethasone premedication, the dose of dexamethasone should be added to line 44 in the abstract.

We agree with the reviewer and have added the dose of premedication to the abstract.

4) Reviewer #3's concern in point #2 is relevant, and the authors' response is appropriate and

helpful. I suggest adding both the question and the response to the discussion section of the manuscript.

We thank the reviewer for this suggestion – we have adapted the discussion section accordingly.